# Wastewater surveillance as a predictive tool for COVID-19: A case study in Chengdu

**Dan Kuang, Xufang Gao, Nan Du, Jiaqi Huang, Yingxu Dai, Zhenhua Chen, Yao Wang, Cheng Wang\*, Rong Lu\***

Department of Environmental and School Health, Chengdu Center for Disease Control and Prevention, Chengdu, Sichuan, China

☙ These authors contributed equally to this work.
\* deerong@126.com (RL); 421093@qq.com (CW)

## Abstract

### Objective

This study was conducted to enhance conventional epidemiological surveillance by implementing city-wide wastewater monitoring of SARS-CoV-2 RNA. The research aimed to develop a quantitative model for estimating infection rates and to compare these predictions with clinical case data. Furthermore, this wastewater surveillance was utilized as an early warning system for potential COVID-19 outbreaks during a large international event, the Chengdu 2023 FISU Games.

### Methods

This study employed wastewater based epidemiology (WBE), utilizing samples collected twice a week from nine wastewater treatment plants that serve 66.1% of Chengdu's residents, totaling 15.2 million people. The samples were collected between January 18, 2023, and June 15, 2023, and were tested for SARS-CoV-2 RNA. A model employed back-calculation of SARS-CoV-2 infections by integrating wastewater viral load measurements with human fecal and urinary shedding rates, as well as population size estimates derived from NH4-N concentrations, utilizing Monte Carlo simulations to quantify uncertainty. The model's predictions compared with the number of registered cases identified by the Nucleic Acid Testing Platform of Chengdu during the same period. Additionally, we conducted sampling from two manholes in the wastewater pipeline, which encompassed all residents of the Chengdu 2023 FISU World University Games village, and tested for SARS-CoV-2 RNA. We also gathered data on COVID-19 cases from the symptom monitoring system between July 20 and August 11.

### Results

From the third week to the twenty-fourth week of 2023, the weekly median concentration of SARS-CoV-2 RNA fluctuated, starting at 16.94 copies/ml in the third week,

**Data availability statement:** All relevant data are within the manuscript and its Supporting information files.

**Funding:** The author(s) received no specific funding for this work.

**Competing interests:** The authors have declared that no competing interests exist.

decreasing to 1.62 copies/ml by the fifteenth week, then gradually rising to a peak of 41.27 copies/ml in the twentieth week, before ultimately declining to 8.74 copies/ml by the twenty-fourth week. During this period, the number of weekly new cases exhibited a similar trend, and the results indicated a significant correlation between the viral concentration and the number of weekly new cases (spearman's r = 0.93, P < 0.001). The quantitative wastewater surveillance model estimated that approximately 2,258,245 individuals (P5-P95: 847,869 - 3,928,127) potentially contracted COVID-19 during the epidemic wave from March 4th to June 15th, which is roughly 33 times the number of registered cases (68,190 cases) reported on the Nucleic Acid Testing Platform. Furthermore, the infection rates of SARS-CoV-2, as estimated by the model, ranged from 0.012% (P5-P95: 0.004% - 0.020%) at the lowest baseline to 3.27% (P5-P95: 1.23% - 5.69%) at the peak of the epidemic, with 15.1% (P5-P95: 5.65% - 26.2%) of individuals infected during the epidemic wave between March 4th and June 15th. Additionally, we did not observe any COVID-19 outbreaks or cluster infections at the Chengdu 2023 FISU World University Games village, and there was no significant difference in the concentrations of SARS-CoV-2 in athletes before and after check-in at the village.

## Conclusions

This study demonstrates the effectiveness of wastewater surveillance as a long-term sentinel approach for monitoring SARS-CoV-2 and providing early warnings for COVID-19 outbreaks during large international events. This method significantly enhances traditional epidemiological surveillance. The quantitative wastewater surveillance model offers a reliable means of estimating the number of infected individuals, which can be instrumental in informing policy decisions.

## Introduction

The global pandemic of severe acute respiratory syndrome coronavirus 2 (SARS-CoV-2) has significantly impacted human life and health, as well as economic and social development, marking it as the most serious public health emergency in recent decades [1–3]. As of May 7, 2023, over 770 million cases of COVID-19 have been reported worldwide, resulting in more than 6.9 million cumulative death [4]. China has endured six waves of the global epidemic, transitioning from acute prevention and control measures to managing the virus as a Class B infectious disease under Category B management, successfully emerging from the pandemic [5]. On May 5, 2023, the World Health Organization declared that COVID-19 no longer constitutes a public health emergency of international concern [6]. It does not imply that COVID-19 is no longer a global health threat. Currently, the novel coronavirus continues to mutate, and the risks associated with the disease persist [7,8]. Additionally, conventional sentinel surveillance, which relied on clinical testing and hospital reporting, faced limitations in capturing true infection dynamics, particularly under reduced testing

frequency and asymptomatic transmission. It is crucial to utilize wastewater-based epidemiology (WBE) as a vital complementary tool to maintain and enhance epidemic surveillance, effectively monitoring virus mutations and the progression of the epidemic.

In the context of COVID-19 emergency prevention and control, China primarily employs a "nucleic acid testing with antigen screening" to monitor COVID-19 infections and provide early warnings of potential outbreaks [9,10]. While this model demonstrates high sensitivity and specificity, it also has significant drawbacks, including the substantial organizational and economic costs associated with large-scale active screening of the population [11,12]. Additionally, there is a risk of missed screenings and population gatherings, which could contribute to the further spread of the epidemic. In the context of a Class B infectious disease under Category B management, the epidemic situation of novel coronavirus infection is primarily monitored through hospitals, communities, and sentinel sites targeting high-risk populations. However, a significant limitation of this approach is that it may not accurately reflect the overall infection status of the population or the dynamics of disease transmission. For instance, hospital sentinel monitoring may overestimate the population's infection rate. Therefore, it is essential to establish and develop a new real-time, efficient, flexible, and cost-effective surveillance technology to supplement conventional sentinel surveillance. WBE is a monitoring method that provides qualitative and quantitative data on population activity or health status within a given area by analyzing the concentration of target chemicals or biomarkers in sewage [13]. WBE can overcome the limitations of traditional methods, such as the need for resource-intensive testing campaigns, the underreporting of asymptomatic or mild cases, and the risks of transmission associated with mass gatherings. It achieves this by providing real-time, population-level data without requiring individual participation. By analyzing SARS-CoV-2 RNA concentrations in sewage, WBE facilitates continuous monitoring of infection trends across entire communities, including unreported cases. This approach offers a more comprehensive and unbiased assessment of epidemic progression [13–15]. For example, the sewage surveillance programme in Hong Kong applied city-wide full-scale interactive methods to assist real-time COVID-19 pandemic control [16].

As a national central city, a highly densely populated mega-city, and an international transportation hub with frequent traffic, it was required to establish a new monitoring and early warning technology platform during the ongoing prevention and control phase of COVID-19. The specific objectives of this study were threefold: (1) To establish a city-wide wastewater surveillance system in Chengdu as a complementary tool to traditional epidemiological monitoring, enabling real-time tracking of SARS-CoV-2 RNA dynamics across nine wastewater treatment plants serving 15.2 million residents. (2) To develop a quantitative model correlating wastewater viral concentrations with clinical case data, aiming to estimate the true scale of COVID-19 infections, including unreported cases, and to evaluate discrepancies between wastewater-based predictions and official nucleic acid testing records. (3) To implement targeted wastewater surveillance in the Chengdu 2023 FISU World University Games village as an early warning system for potential outbreaks, ensuring the timely detection of outbreaks during the international event.

## Materials and methods

### Wastewater treatment plants (WWTPs) selection

This study was conducted with approval from the Chengdu Center for Disease Control and Prevention. No endangered or protected species or human subjects were involved in this research. The selection criteria for the WWTPs were based on the following factors: 1) serving high-density urban populations; 2) encompassing diverse community typologies, including residential areas, commercial centers, academic institutions, and transit hubs; 3) featuring standardized inlet structures, automated sampling systems, and centralized locations within a one-hour transport radius from our laboratory; and 4) having participated in municipal environmental monitoring programs and being able to provide datasets on daily flow rates and NH4-N concentrations. Consequently, nine WWTPs serving the entire population of Chengdu's main urban area (15.2 million residents, which represents 66.1% of the city's total population) were selected for sewage sample collection. Moreover, urban WWTPs in Chengdu generally serve a radius of 5–15 kilometers, with an average of 8 kilometers. The

 

average hydraulic retention time (HRT) from residential areas to WWTPs is influenced by pipe flow velocity and the complexity of the network. Municipal sewage systems in China typically operate at a flow velocity of 1.0 m/s under gravity flow conditions. For an average 8-kilometer pipeline, the HRT is approximately 2.22 hours.

## Samples collection

Sampling began on January 18, 2023, for the 4th, 10th-1, and 10th-2 WWTPs, and on February 20 for the remaining facilities. Twice-weekly collections continued consistently until June 15, 2023. A composite sewage sample of 2.4L was collected from the influent channels of each WWTP at one-hour intervals over a 24-hour period, from 10:00 AM to 10:00 AM the following day, during each sampling occasion. The auto-samplers employed peristaltic pumps to automatically collect 100 mL sewage samples from the sewers at predetermined times and intervals. All collected sewage samples were stored in a refrigerator at 4 °C immediately after collection and were delivered to the laboratories for analysis before 11:00 AM the next day. The sampling frequency for all WWTPs was twice per week, specifically on Mondays and Thursdays, which allows for comprehensive coverage of both weekends and weekdays, thereby enhancing the scientific accuracy of detecting trends in virus concentrations. Additionally, the daily flow of the WWTPs on the sampling dates and the NH4-N concentration of the sewage samples were recorded. Furthermore, we conducted sampling using the same methods at two manholes in the wastewater pipeline, which served the population of the Chengdu 2023 FISU World University Games village. Additionally, daily new registered cases were obtained from the Nucleic Acid Testing Platform of Chengdu.

## Pretreatment of the samples

Upon arriving at the laboratory, sewage samples were bathed in water at 60 °C for 30 minutes, with mixing every 10 minutes to ensure the inactivation of the virus. RNA extraction was performed as described in previous study [17]. Briefly, the sewage samples (35 mL each, three samples) were first centrifuged at 2,500 g for 30 minutes at 4 °C (X4R Pro Centrifuge, Thermo Scientific, Germany). The supernatant was then mixed with 3.5 g ± 0.1 g of polyethylene glycol 8000 (10% w/v, molecular biology grade, Shenggong, Shanghai, China) and 0.79 g ± 0.01 g of sodium chloride (NaCl, high purity grade, Shenggong, Shanghai, China). Subsequently, 35 mL of the supernatant was concentrated through ultracentrifugation at 12,000 g for 120 minutes at 4 °C (X4R Pro Centrifuge, Thermo Scientific, DE). The pellet obtained from the 35 mL sewage was resuspended in approximately 0.4 mL of nuclease-free water and transferred into a new 1.5 mL microcentrifuge tube for further real-time fluorescent reverse transcription polymerase chain reaction (RT-PCR).

## RT-qPCR

0.2 mL of the enriched concentrate was subjected to fully automated nucleic acid extraction. One-step RT-qPCR targeting SARS-CoV-2 open reading frame 1ab, (ORF1ab) and nucleocapsid protein (N) was used for the detection and quantification of virus RNA concentrations with the same reagents and recommended annealing temperatures refer to the kit instructions (SARS-CoV-2 nucleic acid detection kit, Liferiver Bio-Tech, Shanghai, PRC). In detail, a 5μL RNA template was used for running 45 cycles in 20μL reaction mixture on the Real-Time PCR Systems (QuantStudio Dx, Thermo Fisher, SG). Diethylpyrocarbonate (DEPC) treated water as negative template controls (NTCs), and DNA plasmids as positive control were included for every batch, and each sample and control were detected in triplicate, with intra-assay coefficients of variation for Ct values ≤ 5%. A sewage sample would be classified as "positive" only if the Ct values of either the ORF1ab or N genes were below 43, and its virus concentration was calculated according to the plate-specific standard curve ($R^2 > 0.99$, efficiency: 90%–110%) established by ten-fold serial dilution of the reference materials (GBW(E)091133, Bdsbiotech, Guangzhou, PRC) from $10^6$ to $10^1$ copies/ml. A sewage sample's corresponding viral concentration was defined as the N1 gene concentrations. The weighted average viral load in wastewater in Chengdu was calculated using the following equation.

$$\text{Weighted average viral load}_{\text{city}} = \frac{\sum_{i=1}^{9} Cn \times Qn}{\sum_{i=1}^{9} Qn} \tag{1}$$

Where Cn represents the concentration of SARS-CoV-2 RNA in wastewater at WWTPn (copies/mL), and Qn represents the daily 24-hour volume of wastewater at WWTPn (mL/day).

**Back-calculation of SARS-CoV-2 infected cases and infection rates based on WBE**

The stool and urine of infected persons are the main contributors to the viral load in wastewater. The amount of SARS-Cov-2 RNA shedding in the stool of an infected person was assumed $1.7*10^6$ according to $10^{2.7}$–$10^{7.8}$ copies/mL reported in previous studies [18]. The daily stool mass per person was assumed 211 g as reported by Rose et al [19].The viral load in urine followed as distribution with a mean of 2.91 $\log_{10}$ copies/ml in urine and an average urination amount of 1500 mL. Daily SARS-Cov-2 viral load in the samples was calculated by the detected virus concentration multiplied by the daily flow rates. Then, the predicted number of persons infected was calculated by the following equation according to the above-assumed viral load in stool and urine.

$$\text{Persons Infected} = \frac{Cr * Q}{Cf*211 + (Cru) * 1.5} \tag{2}$$

Where Cr represents the concentration of SARS-CoV-2 RNA in wastewater (copies/L), Q represents the daily 24-hour volume of wastewater at WWTPn (L/day), Cf represents the concentration of SARS-CoV-2 RNA in feces (copies/g), and Cru represents the concentration of SARS-CoV-2 RNA in urine (copies/L).

The population size in the catchment area of the WWTP was calculated according to the concentration of NH4-N in wastewater. Previous studies reported that the amount of ammonium nitrogen (NH4-N) emission in the wastewater of one person was 4.2 g per day [13,20]. Thus, the population size of WWTP's coverage area was calculated by the following equation.

$$\text{Population size} = \frac{CN * Q}{4.2\text{g/person} \cdot \text{day}} \tag{3}$$

Where CN represents the concentration of NH4-N in wastewater (g/L), and Q represents the daily 24-hour volume of wastewater at WWTPn (L/day).

The Monte Carol simulation was carried out in Excel (Microsoft, Redmond, WA) with ModelRisk version 6.0. The number of iterations per simulation was 10,000. The predicted infection cases are reported as the median and 95% confidence interval (CI) determined by bootstrapping the model with 100 experiments of 1000 draws each.

The infection rate of SARS-CoV-1 (Pi) was calculated as follows:

$$Pi\,(\%) = \frac{\sum_{i=1}^{9} ni}{\sum_{i=1}^{9} Ni} \tag{4}$$

where ni are the predicted numbers of infected individuals in the catchments of the nine WWTPs in each district, Ni are the population size in the catchments of the nine WWTPs in each district.

**Data analysis**

Spearman's correlation was conducted to examine the relationship between weekly viral load in wastewater and the number of registered cases. Statistical analysis was performed using R software version 4.3.2.

## Results

### The characteristics of nine WWTPs

Wastewater samples were collected from nine WWTPs (numbered 3–10 according to Chengdu's administrative codes). The 9th WWTP exhibited the highest average daily flow (996.7 thousand m³/d), while the subdivided 10th-1 and 10th-2 WWTPs had flows of 48.1 and 249.2 thousand m³/d, respectively. The median concentration of NH4-N varied between 20.1 and 40.8 mg/L. Collectively, these nine WWTPs serve a population of approximately 15.2 million, based on the daily flow and NH4-N concentration, which accounts for 66.1% of the total residents of Chengdu. The wastewater samples from the 4th, 10th-1, and 10th-2 WWTPs were collected on January 18, 2023, while the other WWTPs collected their samples on February 20, 2023 (Table 1).

### Epidemic trend in Chengdu

During the study period from January 18 to June 15, 2023, the number of registered cases in the Nucleic Acid Testing Platform of Chengdu gradually decreased from 1,343 cases per day on January 6–14 cases per day on April 9. Subsequently, the number began to rise again, peaking at 3,220 cases in mid-May, before returning to a relatively

**Table 1. The characteristics of nine WWTPs.**

| WWTP | Catchment Zone | Average Daily Flow (10,000 m³/d) | temperature (°C, P5-P95) | Total Suspended Solids (mg/L, P5-P95) | Chemical Oxygen Demand (mg/L, P5-P95) | NH4-N (mg/L,P5-P95) | Population (P5-P95) | Initial Sampling Date | Sampling Frequency | Samples Number |
|---|---|---|---|---|---|---|---|---|---|---|
| the 3rd | High-tech | 16.14 | 20.0 (9.8-25.4) | 291.5 (173.0-993.0) | 553.5 (296.3-1272.5) | 35.1 (29.3-41.1) | 1,334,750 (1,127,976-1,741,667) | 20-Feb | twice/week | 34 |
| the 4th | Chenghua | 10.53 | 20.8 (11.3-23.9) | 185.0 (85.8-363.8) | 422.0 (318.2-616.0) | 40.8 (33.5-46.8) | 1,004,160 (782,930-1,227,782) | 18-Jan | twice/week | 43 |
| the 5th | Wuhou | 14.69 | 20.0 (15.8-22.0) | 124.5 (61.5-263.8) | 346.5 (255.8-574.5) | 36.9 (31.9-42) | 1,278,834 (1,045,587-1,542,056) | 20-Feb | twice/week | 34 |
| the 6th | Chenghua | 7.31 | 20.4 (12.9-23.8) | 155.0 (79.0-331.8) | 363.0 (207.3-576.3) | 37.7 (24.3-46.6) | 659,273 (421,380-800,207) | 20-Feb | twice/week | 34 |
| the 7th | Jinniu | 10.06 | 20.5 (16.7-22.8) | 365.7 (145.0-708.0) | 284.6 (126.4-613.6) | 20.1 (5.8-29.8) | 507,858 (126,332-694,306) | 20-Feb | twice/week | 34 |
| the 8th | Qingyang | 16.31 | 19.5 (16.3-22.6) | 125.5 (71.8-276.3) | 317.5 (187.0-925.0) | 32.5 (22.1-38.6) | 1,252,835 (837,191-1,530,330) | 20-Feb | twice/week | 34 |
| the 9th | Jinjiagn | 99.67 | 20.9 (16.6-23.4) | 222.0 (61.3-423.5) | 363.0 (207.8-590.3) | 28.3 (25.8-32) | 6,750,028 (6,145,291-7,698,806) | 20-Feb | twice/week | 34 |
| the 10th-1 | High-tech | 4.81 | 17.0 (8.0-23.8) | 89.0 (50.0-202.8) | 249.0 (140.2-400.0) | 36.1 (24.1-44) | 419,229 (246,535-513,853) | 18-Jan | twice/week | 43 |
| the 10th-2 | High-tech | 24.92 | 18.0 (8.0-23.0) | 160.0 (56.2-288.2) | 346.0 (166.2-529.2) | 32.8 (27-37.5) | 1,959,031 (1,338,667-2,435,584) | 18-Jan | twice/week | 43 |

Note: Initial sampling dates indicate the first collection day for each WWTP; sampling continued twice weekly until June 15, 2023. The total sample numbers reflect cumulative collections over the entire study period.

low level by mid-June 2023 (Fig 1). Wastewater surveillance monitoring was conducted from January 18 to June 15, 2023, using influent samples from nine WWTPs of Chengdu, which captured the unimodal upward and downward trends of registered cases. During the study period, 97.0% of sewage samples tested positive, maintaining a 100% positivity rate from the third week to the twelfth week, before dropping to 83.3% in the fourteenth week, and then progressively climbing back to 100% by the end of the twenty-first week. From the third week to the twenty-fourth week of 2023, the weekly median concentration of SARS-CoV-2 RNA started at 16.94 copies/ml in the third week and gradually decreased to 1.62 copies/ml by the fifteenth week. An increase in viral concentration was observed from late April until mid-June, reaching a maximum median concentration of 41.27 copies/ml. This peak was followed by a rapid decline, with the median concentration falling below 10 copies/ml by mid-June (Fig 2). Throughout this period, weekly virus concentrations were significantly correlated with the weekly number of new cases (spearman's r = 0.93, P < 0.001) (Fig 3).

**Prediction of the actual infection cases and rates by the quantitative wastewater surveillance model**

The quantitative wastewater surveillance model and Monte Carlo simulation were employed to estimate the actual number of infection cases on each sampling day. On April 13th, which recorded the lowest viral concentration, a median of 6,517 infected individuals was estimated (P5-P95: 2,447–11,337). The estimated number of infected cases rose to 18,483 (P5-P95: 6,939–32,148) on April 27th, just before May Day, and further increased to 60,767 (P5-P95: 22,812–105,696) on May 4th, the first day of May Day. The maximum median number of infected cases reached 486,384 (P5-P95: 182,650–846,087) on March 18th. This indicates that approximately 2,258,245 (P5-P95: 847,869-3,928,127) individuals potentially contracted COVID-19 during the epidemic wave between March 4th and June 15th, which is about 33 times the number of registered cases (68,190 cases) reported on the Nucleic Acid Testing Platform. Furthermore, the infection rates of SARS-CoV-2, as estimated by the model, ranged from 0.012% (P5-P95: 0.004%-0.020%) at the baseline to 3.27% (P5-P95: 1.23%-5.69%) at the peak of the epidemic, with 15.1% (P5-P95: 5.65%-26.2%) of the population infected during the epidemic wave between May 4th and June 15th (Fig 4).

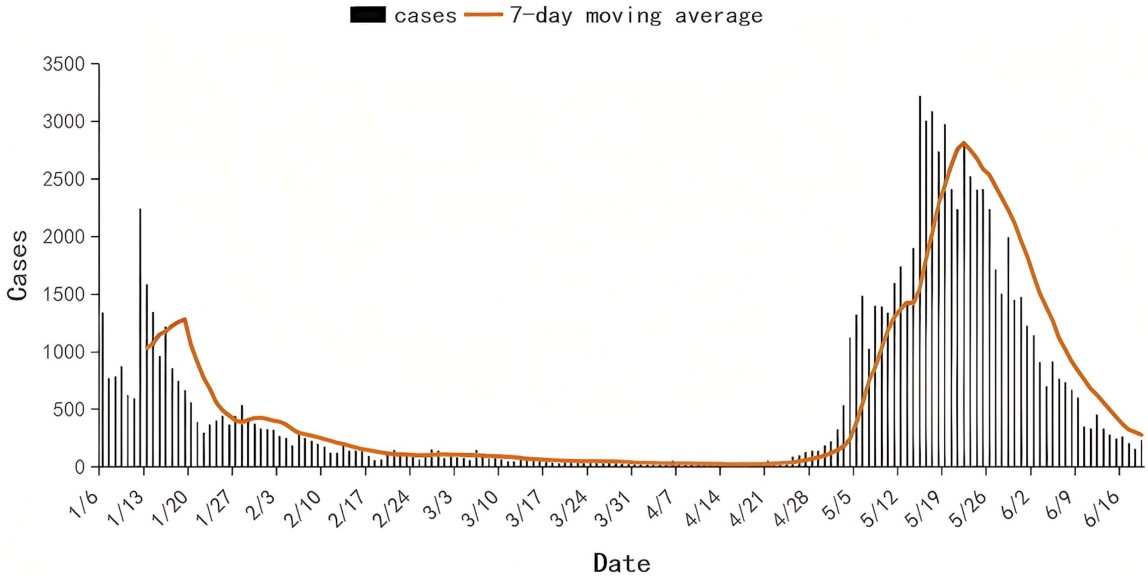

**Fig 1. The number of registered COVID-19 cases per day in Chengdu from from 18 January to 15 June 2023.**

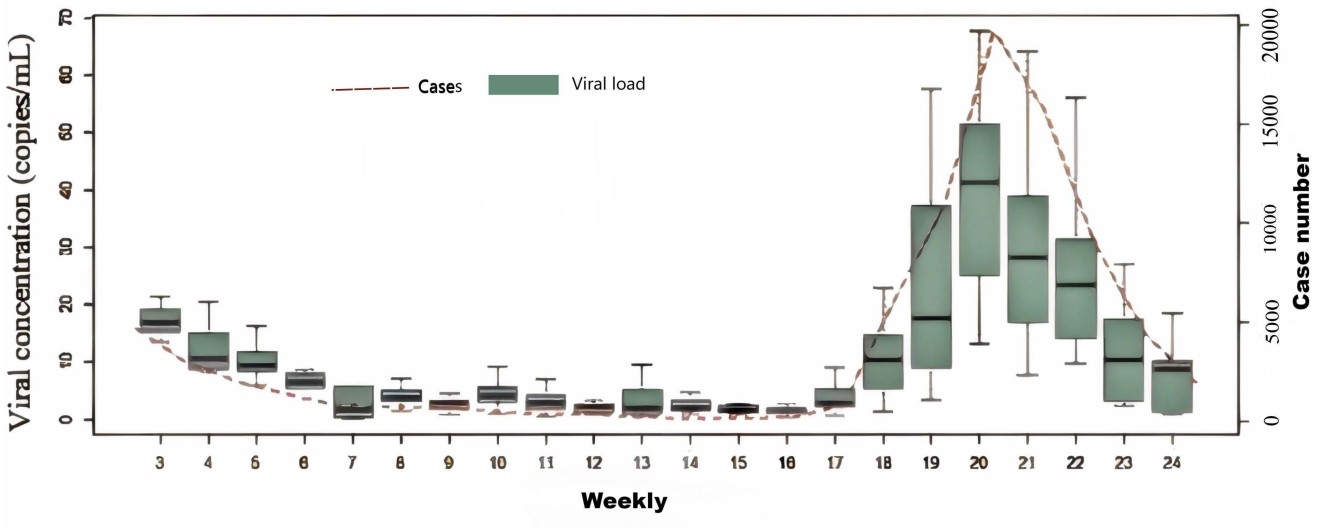

**Fig 2. The viral load of SARS-CoV-2 in wastewater and weekly reported cases in Chengdu from 18 January to 15 June 2023.**

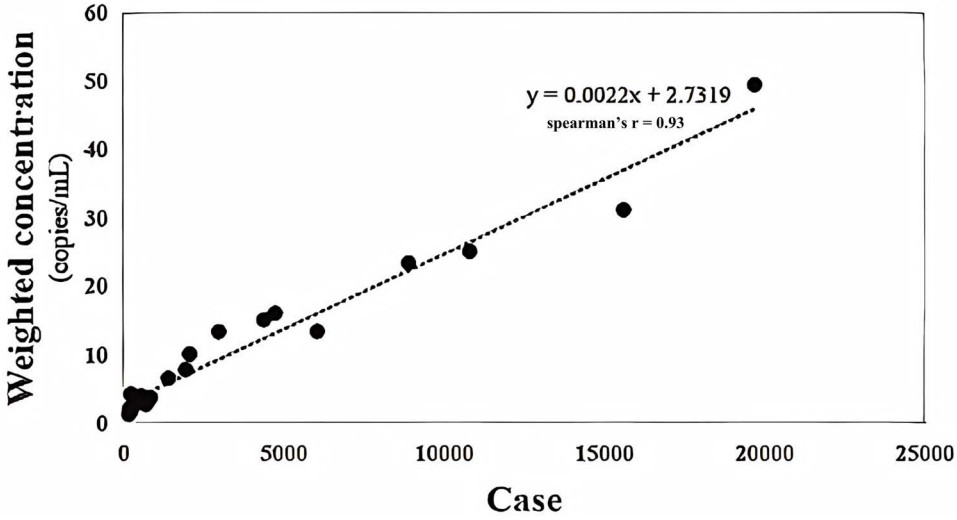

**Fig 3. The correlation between the viral loads of SARS-CoV-2 in wastewater and the weekly reported cases.**

## Application of wastewater surveillance for early warning of COVID-19 outbreak in Chengdu 2023 FISU world university games village

As an early warning system for the COVID-19 outbreak at the Chengdu 2023 FISU World University Games Village, we conducted sampling in two manholes of the wastewater pipeline, which served the entire population of the area. The first SARS-CoV-2-positive water sample was detected on June 29th, with an average virus concentration of 0.56 copies/mL (standard error (se) = 0.37). The concentration of SARS-CoV-2 decreased to 0.45 copies/mL (se = 0.45) on July 3th, 0.48 copies/mL (se = 0.48) on July 10th, and further declined to undetectable levels on July 17th. Athletes began checking in at the Chengdu 2023 FISU World University Games village on July 20th, and we conducted symptom monitoring and

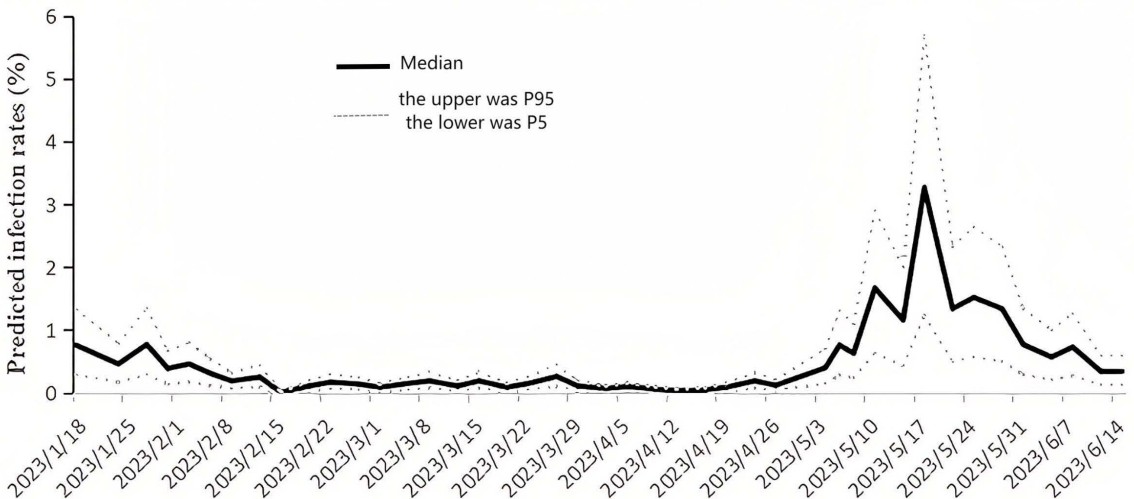

**Fig 4. Predicted infection rate from 18 January to 15 June 2023.** Predicted values of infection rate represented the median values of Monte Carol Simulation. P5-P95 is shown in grey.

wastewater surveillance simultaneously. Between July 20th and August 11th, during which the Chengdu 2023 FISU World University Games village officially opened and closed, 23 COVID-19 cases were reported by the symptom monitoring system; however, we did not observe any COVID-19 outbreaks or cluster infections. There were five days during which SARS-CoV-2 was not detected in the wastewater, and the concentration of SARS-CoV-2 reached two peaks on August 4th (mean: 0.43 copies/mL, se: 0.03) and August 9th (mean: 0.24 copies/mL, se: 0.01) (Fig 5). There was no significant difference in the concentrations of SARS-CoV-2 in athletes before (mean: 0.37 copies/mL, se: 0.13) and after check-in (mean: 0.08 copies/mL, se: 0.03) at the Chengdu 2023 FISU World University Games village (U = 15.5 P = 0.08).

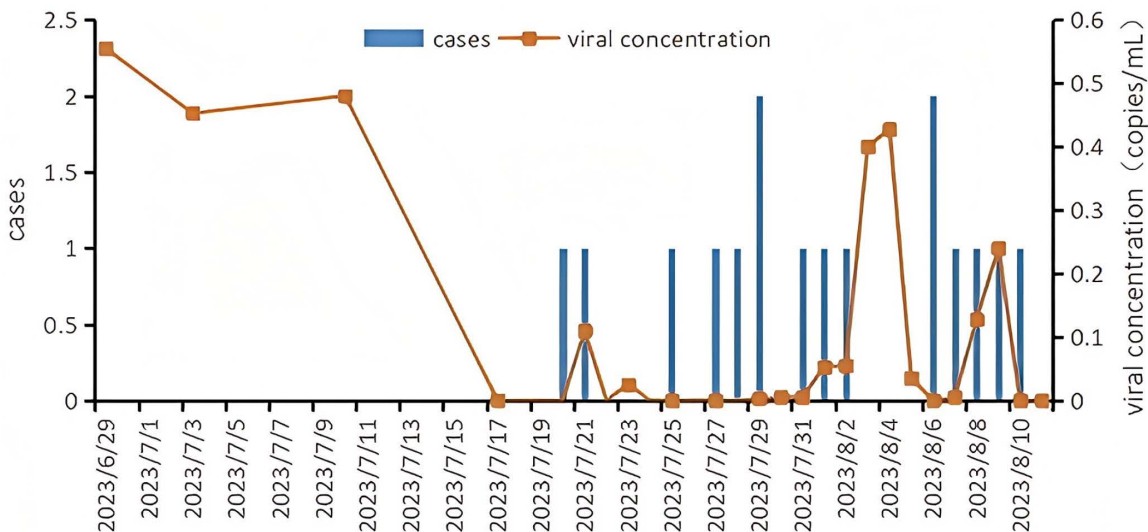

**Fig 5. The viral load of SARS-CoV-2 in wastewater and the reported cases in Chengdu 2023 FISU world university games village.**

## Discussion

This study presents an initial report on the application of WBE for the surveillance of SARS-CoV-2 in the post-epidemic era in Chengdu and the Chengdu 2023 FISU World University Games village. Both the epidemiological surveillance and wastewater data revealed an epidemic wave following the lifting of strict COVID-19 measures in Chengdu after five months, with the cumulative infection rate rising from an extremely low level to 15.1%. The wastewater data indicated that there was no COVID-19 outbreak or cluster infection in the Chengdu 2023 FISU World University Games village. Wastewater surveillance during a large international gathering was more effective and enhanced traditional monitoring methods.

Our results show that a second epidemic wave occurred five months after the first wave, following the policy adjustment. The peak SARS-CoV-2 viral load detected in wastewater in Chengdu (4.62 log10 copies/L) was significantly lower than the viral loads observed in Shijiazhuang (7.38 log10 copies/L) [21] and Shenzhen (7.15 log10 copies/L) [22], which corresponded to the first epidemic wave after the strict COVID-19 measures were lifted. However, the viral load levels in our study were comparable to those reported in the Philippines (4.9 log10 copies/L) [23], Brazil (4.78 log10 copies/L) [24], and Japan (4.88 log10 copies/L) [25]. This discrepancy may be attributed to the fact that over 80% of the susceptible Chinese population had been infected with SARS-CoV-2, leading to increased viral shedding in wastewater several weeks after the lifting of strict COVID-19 measures in December 2022. Consequently, the population developed a high degree of natural immunity, resulting in fewer individuals being infected and shedding the virus at baseline. As antibody levels declined post-infection and fell below the threshold for protection, a second epidemic wave emerged. Fortunately, the intensity of this wave was lower than that of the initial epidemic wave, which aligns with the experiences of other countries that have faced multiple waves of SARS-CoV-2.

Our study demonstrated a significantly strong correlation coefficient (spearman's r = 0.93) between weekly virus concentrations and COVID-19 case numbers. Previous studies have also indicated a strong correlation between these results, while other research reported lower correlation coefficients ranging from 0.19 to 0.53 [26–28]. One possible reason for this discrepancy may be the choice of primer and probe sets; higher correlation coefficients were observed when detecting viral RNA in wastewater and populations using the same N1 and/or N2 primer and probe sets. Additionally, some studies focused on active or cumulative cases reported in Canada [28], Saudi Arabia [29], and the USA [30], resulting in correlation coefficients between 0.21 and 0.51. A meta-analysis further indicated that the correlation coefficient between WBE data and clinically identified cases was higher for new cases compared to active cases [31]. These results suggest that WBE is more effective in capturing new cases than in tracking active and cumulative cases, likely due to higher shedding loads from newly infected patients. Furthermore, previous studies have revealed that other factors, such as wastewater temperature, catchment size, chemical dosing, and the accuracy of clinically confirmed cases, also influence correlation coefficients [31]. Higher wastewater temperatures and larger catchment sizes were associated with lower correlation coefficients, likely due to increased decay of SARS-CoV-2 RNA [32,33]. The correlation between WBE data and clinically confirmed cases may be also biased by the extent to which reported cases reflect true infections [31].

After COVID-19 was classified under Category B management, numerous symptomatic and asymptomatic patients, whether tested or not, went unreported. Therefore, it is crucial to predict the actual number of infected cases and assess the trends of the epidemic. The back-calculation of infected cases by WBE-based prediction model, which incorporated dynamic viral shedding, indicated that 3.27% of the population was infected at the peak of the second wave following policy adjustments in Chengdu. The estimated peak infection burden from wastewater was approximately 33 times higher than the officially reported cases during the same period. This substantial disparity strongly indicates significant underreporting following China's transition to Category B management, which excluded both untested symptomatic individuals and asymptomatic carriers from surveillance systems. This underdetection aligns with global observations that wastewater surveillance captures 11 times more COVID-19 cases than clinical reporting [34], underscoring the critical role of wastewater epidemiology in complementing conventional surveillance. The results are consistent with monitoring data from national sentinel hospitals, which indicated that the positive rate of COVID-19 among outpatients and emergency

department cases was 2.5% (the proportion of influenza-like illness cases was 6.2%, and the positive rate of COVID-19 among influenza-like illness cases was 40.7%) [35]. However, these findings are inconsistent with the results from Taiyuan City, which utilized a dynamic model and suggested that the peak of infectious individuals during the second wave was only 0.88% [36]. These discrepancies are likely attributable to the different models used to calculate infected cases and the unique history of infection in each location. The model in our study relies on the following critical assumptions, which are grounded in published literature but inherently introduce uncertainty. First, the fixed fecal and urine shedding rates are derived from literature medians and do not account for individual variability (e.g., differences between asymptomatic and symptomatic shedding, as well as host immunity effects). This oversight could result in either overestimation or underestimation of the true shedding rates within the population. Second, the population served by the WWTP is calculated using NH4-N concentration. However, industrial and commercial NH4-N inputs (e.g., from fertilizers and food processing) or transient populations (e.g., tourists) could inflate population estimates, leading to an underestimation of infection rates. Third, the measured viral concentrations in wastewater directly reflect contributions from infected individuals, without explicit adjustments for RNA degradation or in-sewer dilution. RNA degradation may lead to an underestimation of true viral loads, particularly during warmer seasons or in areas with long hydraulic retention times. Additionally, dilution from stormwater inflow could further reduce measured concentrations, biasing infection estimates downward. Numerous studies have struggled to achieve accurate and reliable back-calculation using WBE, including those that utilized machine learning models, developed artificial neural network models, and applied artificial intelligence [37–39]. However, few studies have employed WBE back-calculation to provide infection case numbers during the second wave after the strict COVID-19 strategies were lifted in China.

In order to ensure the stability of the study results, we conducted sensitivity analyses. The faecal SARS-CoV-2 RNA shedding rate was chosen to be 82.4%; the daily faecal production per person was 128g, and the faecal NH4-N excretion varied ±10% from baseline values. The predicted infection rates calculated according to different parameters were similar (S1 Tables–S3). Therefore, the results calculated by the model were reliable.

Our study provides valuable insights. First, the quantitative model's ability to estimate infection rates and hidden transmission. Second, the novel application of WBE for preemptive outbreak prevention at large international events. There were also several limitations. First, we reported that the viral loads in wastewater during the second wave were significantly lower than those in the first wave after the policy adjustments. However, sewage samples were not collected or analyzed during the first wave in this study. Consequently, we could only compare our findings with viral load data from the literature regarding the first wave. Second, while we obtained data on daily new registered cases, our sampling frequency was limited to twice per week. This limitation hindered our ability to demonstrate a clear relationship between the adjusted daily viral load and the number of registered cases, potentially resulting in a loss of information and reduced statistical performance. Third, our back-calculation model did not account for the decay rate of the virus in sewers, which is influenced by wastewater temperature and the distance from the community to the WWTP. This oversight may have led to an underestimation of the viral load in wastewater and the actual number of infected individuals. Furthermore, the fecal and urine viral shedding loads in our model appear to vary between individuals and infection states, resulting in significant variability in the results. Further optimization of the modeling approach is necessary to enhance the accuracy of our findings. Fourth, we did not analyze the gene sequencing of the wastewater samples due to the limited number of samples and their low quality. In the future, we aim to improve our methods for analyzing low-quality sequence data and estimating the relative abundance of lineages in wastewater, thereby facilitating the early detection of new variants.

## Conclusions

This study establishes WBE as a crucial tool for tracking SARS-CoV-2 during Chengdu's post-pandemic phase. Monitoring nine wastewater treatment plants that serve 15.2 million residents revealed a second epidemic wave five months after the lifting of restrictions, which strongly correlated with clinical trends. Model estimates indicated approximately 2.26

million infections, 33 times higher than the reported cases, peaking at an infection rate of 3.27%. WBE provided an early warning at the FISU Games Village, aligning with symptom monitoring efforts. Despite the limitations of sampling and back-calculation models, including fecal and urine viral shedding rates, RNA degradation, and in-sewer dilution, WBE has proven to be effective in identifying hidden transmission and guiding public health responses.

## Supporting information

**S1 Table. Summary of correlations in previous studies.**
(XLSX)

**S2 Table. The raw data needed to replicate the findings of this study.**
(XLSX)

**S3 Table. The results in different models.**
(DOCX)

## Acknowledgments

The authors are grateful to the colleagues who participated in epidemiological investigations, sample collection, laboratory testing, and data analysis.

## Author contributions

**Data curation:** Dan Kuang, Nan Du, Jiaqi Huang, Yao Wang.

**Formal analysis:** Dan Kuang.

**Investigation:** Nan Du, Xufang Gao, Yingxu Dai.

**Methodology:** Dan Kuang, Zhenhua Chen.

**Writing – original draft:** Dan Kuang, Jiaqi Huang.

**Writing – review & editing:** Dan Kuang, Cheng Wang, Rong Lu, Xufang Gao, Yingxu Dai.

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
