## [Decision Letter · Decision Letter 0]

17 Mar 2025

PONE-D-25-06286Use of wastewater surveillance for COVID-19: a practice in ChengduPLOS ONE

Dear Dr. Lu,

Thank you for submitting your manuscript to PLOS ONE. After careful consideration, we feel that it has merit but does not fully meet PLOS ONE’s publication criteria as it currently stands. Therefore, we invite you to submit a revised version of the manuscript that addresses the points raised during the review process.

Regarding the reviewers' comments, we came to conclusion that the manuscript requires MAJOR revision. Authors are strongly recommended to consider the whole comments of reviewers carefully. Authors must properly revise/ respond to those comments that need more explanations, require reasons or discussions, and clarifications of the originality or data analysis and validation. These queries are more or less found among the comments of all three reviewers and are critical for any future acceptance.

We look forward to receiving your revised manuscript.

Kind regards,

Shervin Jamshidi

Academic Editor

PLOS ONE

Journal Requirements:

4. Please ensure that you refer to Figure 5 in your text as, if accepted, production will need this reference to link the reader to the figure.

Reviewers' comments:

Reviewer's Responses to Questions

**Comments to the Author**

1. Is the manuscript technically sound, and do the data support the conclusions?

Reviewer #1: Yes

Reviewer #2: Yes

Reviewer #3: Yes

2. Has the statistical analysis been performed appropriately and rigorously? 

Reviewer #1: Yes

Reviewer #2: No

Reviewer #3: Yes

3. Have the authors made all data underlying the findings in their manuscript fully available?

Reviewer #1: No

Reviewer #2: Yes

Reviewer #3: Yes

4. Is the manuscript presented in an intelligible fashion and written in standard English?

Reviewer #1: Yes

Reviewer #2: Yes

Reviewer #3: Yes

5. Review Comments to the Author

Reviewer #1: 

Add the specific objectives of the study.

This study employed wastewater based epidemiology (WBE), utilizing samples collected twice a week from nine wastewater treatment plants that serve 66.1% of Chengdu's residents, totaling 15.2 million people. A complete paragraph on the choice of stations with a clarification of the sampling plan is necessary. Why not choose a number of stations covering 80% of the population of Chengdu? Explain the details of the criteria for choosing these stations, as the size of the population must be part of a relevant multi-criteria set.

Can you explain why Spearman was chosen instead of Pearson?

Results: From the third week to the twenty-fourth week of 2023, the weekly median concentration of SARS-CoV-2 RNA fluctuated, starting at 16.94 copies/ml in the third week, decreasing to 1.62 copies/ml by the fifteenth week, then gradually rising to a peak of 41.27 copies/ml in the twentieth week, before ultimately declining to 8.74 copies/ml by the twenty-fourth week. In fact, various parameters (flow rate from stations, information on hydraulic management of networks, chemical parameters, etc.) are most often integrated in order to correct these fluctuations as effectively as possible. Why didn't you take into account the data on fluctuation factors? What have you done statistically to smooth out the observed fluctuations?

Reviewer #2: 

It is unclear why samples were taken from specific manholes in addition to WWTPs?

In Abstract, 66% of population is served by nine WWTPs but in Line 106: entire population is served.

It is unclear where samples are collected? the influent or effluent of WWTPs.

Since 2.4-L sample is collected during 24 hours, its biological content is doubtful. Its viral content would be deteriorated from the early auto-sampling. Each sample mainly represents the viral content of late sampling (e.g. 8-10 AM) than its previous day.

Usually, there is a hydraulic retention time between the residential areas and WWTPs in wastewater collection systems. Authors should clarify with proper data how much time it takes on average for the sewage to travel from these areas to 9 WWTPs. It affects both the viral content and early warning systems.

Line 194: It is confusing. authors are pointing to 9 WWTPs, but there is a 10th WWTP. Again in Line 194: The flow of 9th WWTP equals the 10th WWTP: 996.7. This confusion continues in Line 201: 10th-1 and 10th-2 WWTPs?

Line 202 and Table 1: Why the sampling dates are different to the study period? wasn't it presumed that sampling is continued to June 15? Table 1 is confusing in numbering the WWTPs and the time of samples.

How much is it possible that the daily sewage of some individuals are considered in two or more WWTPs? I mean if someone is infected and lives in Place A, but works in Place B, its wastewater would enter two WWTPs. How much is the rate of these individuals?

Line 213-217: how much is the accuracy of experiments? It should be clarified, particularity when you are pointing to small variations (83%-100%).

I assumed that cases point to new daily and weekly cases. But why in Figure 4, the number of cases are steady for some weeks?

Figure lack proper legends.

Authors must clarify their specific innovation of their research, particularly in comparison with similar studies in the same area (like reference No.16)

Line 291: r = 0.98 or r2?

Regarding Line 304-306, it is necessary for authors to report the status of WWTPs.

I wonder how much personal curation would affect the results? I mean people have gradually learned how to cure themselves without going to hospitals for COVID-19. It may significantly change the rates and correlation.

Conclusion needs some major revision to illustrate the whole findings of research.

The whole results require statistical attachments about the average, standard errors, and the significance of difference between outbreaks.

Reviewer #3: 

(Line 1): The title is informative but could be more precise. Consider specifying the study's key objective, such as "Wastewater Surveillance as a Predictive Tool for COVID-19: A Case Study in Chengdu."

(Line 19-21): The abstract mentions a "quantitative wastewater surveillance model," but does not briefly describe how it works. Adding a sentence explaining the core method used (e.g., back-calculation) would improve clarity.

(Line 37-38): The phrase "which is roughly 33 times the number of registered cases" is important but should highlight the implications, which is does this suggest significant underreporting?

(Line 62-66): "It does not imply that COVID-19 is no longer a global health threat." This sentence is clear, but it would be stronger if linked to the need for alternative surveillance systems, reinforcing the relevance of the study.

(Line 75-83): The section discussing China’s monitoring system could be improved by briefly explaining how WBE overcomes the drawbacks of traditional methods.

(Line 85-95): The rationale for WBE is clear, but a brief mention of the key studies supporting its use would strengthen the argument.

(Line 117-118): Consider explaining why NH4-N concentration was chosen to estimate population size.

(Line 145-146): The statement about controls should explicitly state whether negative controls were included.

(Line 148-149): The cut-off Ct value of 43 seems high. Justify why this threshold was used.

(Line 163-164): The assumed stool shedding rate is taken from previous studies but should be validated against recent WBE literature.

(Line 174-178): The use of NH4-N to estimate population size is reasonable but should include a reference for its validity in previous studies.

(Line 224-225): The R² value (0.9633) indicates a strong correlation, but was the analysis adjusted for potential confounders such as changes in testing capacity?

(Line 237-239): The statement that "approximately 2,258,245 individuals contracted COVID-19" is significant. Consider discussing uncertainty in this estimate (e.g., model limitations, underestimation due to viral degradation).

(Line 240-242): The infection rates estimated should be compared with independent estimates from seroprevalence or epidemiological models.

(Line 265-273): The comparison of wastewater data with symptom monitoring results is useful but lacks an explicit statement on whether the wastewater method was more effective.

(Line 274-289): The comparison of Chengdu’s epidemic curve with other regions is insightful, but the interpretation of why Chengdu’s viral loads were lower than in Shijiazhuang and Shenzhen needs more explanation.

(Line 291-303): The discussion on correlation coefficients is strong but would benefit from a brief table summarizing the different correlation values observed in past studies.

(Line 311-319): The back-calculation model should be discussed in more detail. What are its key assumptions? How do those assumptions affect accuracy?

(Line 347-353): The conclusion is strong but should include a brief mention of key limitations, particularly regarding model assumptions.

6. PLOS authors have the option to publish the peer review history of their article (what does this mean? ). If published, this will include your full peer review and any attached files.

**Do you want your identity to be public for this peer review?** For information about this choice, including consent withdrawal, please see our Privacy Policy .

Reviewer #1: No

Reviewer #2: No

Reviewer #3: No

---

## [Author Response · Author response to Decision Letter 1]

2 Apr 2025

Dear Reviewers,

We greatly appreciate your comments and suggestions. In response to the reviewers' feedback, we have addressed each point raised by the academic editor and the reviewers. Additionally, we have made some changes to the manuscript, which are highlighted for your convenience.

Authors Responses to Journal Requirements

AUTHOR RESPONSE: We have thoroughly reviewed the PLOS ONE style templates and confirm that the manuscript now fully complies with all formatting guidelines, including file naming conventions. All sections (title page, main text, references, and figures) have been adjusted to meet the journal's requirements.

AUTHOR RESPONSE: In the Methods section (specifically in the "Samples collection" subsection), we have added the following clarification:

"This study was conducted with approval from the Chengdu Center for Disease Control and Prevention. No endangered or protected species or human subjects were involved in this study." (Line 131-133)

AUTHOR RESPONSE: We confirm our commitment to PLOS ONE's open data policy. Our complete dataset will be deposited in the Supporting information (S2 Table. The raw data needed to replicate the findings of this study) of this article upon acceptance.

4. Please ensure that you refer to Figure 5 in your text as, if accepted, production will need this reference to link the reader to the figure.

AUTHOR RESPONSE: We confirm that Figure 5 is now correctly cited in the text, first appearing in the Results section (specifically in the “Application of wastewater surveillance for early warning of COVID-19 outbreak in Chengdu 2023 FISU world university games village” subsection). (Line 322)

Author’s Responses to Review Comments

Comments from Reviewer #1:

1. Add the specific objectives of the study.

AUTHOR RESPONSE: Many thanks for the Reviewer’s valuable suggestion.We have carefully revised the Abstract and Introduction sections to explicitly state the specific objectives of the study. It is as follows:

Corresponding update in Abstract (Objective section): "This study was conducted to enhance conventional epidemiological surveillance by implementing city-wide wastewater monitoring of SARS-CoV-2 RNA. The research aimed to develop a quantitative model for estimating infection rates and to compare these predictions with clinical case data. Furthermore, this wastewater surveillance was utilized as an early warning system for potential COVID-19 outbreaks during a large international event, the Chengdu 2023 FISU Games)." (Line 21-26)

Revised text in Introduction section: "The specific objectives of this study were threefold: (1) To establish a city-wide wastewater surveillance system in Chengdu as a complementary tool to traditional epidemiological monitoring, enabling real-time tracking of SARS-CoV-2 RNA dynamics across nine wastewater treatment plants serving 15.2 million residents. (2) To develop a quantitative model correlating wastewater viral concentrations with clinical case data, aiming to estimate the true scale of COVID-19 infections, including unreported cases, and to evaluate discrepancies between wastewater-based predictions and official nucleic acid testing records. (3) To implement targeted wastewater surveillance in the Chengdu 2023 FISU World University Games village as an early warning system for potential outbreaks, ensuring the timely detection of outbreaks during the international event." (Line 115-127)

2. This study employed wastewater based epidemiology (WBE), utilizing samples collected twice a week from nine wastewater treatment plants that serve 66.1% of Chengdu's residents, totaling 15.2 million people. A complete paragraph on the choice of stations with a clarification of the sampling plan is necessary. Why not choose a number of stations covering 80% of the population of Chengdu? Explain the details of the criteria for choosing these stations, as the size of the population must be part of a relevant multi-criteria set.

AUTHOR RESPONSE: We sincerely appreciate your constructive feedback on our study design. Below, we provide a detailed clarification regarding the selection criteria for wastewater treatment plants and the rationale for population coverage.

(1)The nine WWTPs were systematically selected based on a hierarchical evaluation of the following criteria:

First, regarding population coverage and density prioritization, the six urban districts covered by these WWTPs (Qingyang, Jinjiang, Wuhou, Jinniu, Chenghua, and the High-Tech Zone) constitute socio-economic core of Chengdu, serving 15.2 million residents (66.1% of the city’s total population). These districts exhibit a population density of 15,854 persons/km², 6.2 times higher than that of suburban areas. High-density urban populations are epidemiologically critical for wastewater-based surveillance, as they enhance community transmission signals and reduce spatial dilution effects.

Second, regarding geographic representativeness, the selected districts include a variety of community typologies, such as residential areas, commercial hubs, academic clusters, and transit centers, reflecting diverse population mobility patterns. This diversity enhances the generalizability of viral trend detection across urban demographics.

Third, concerning infrastructure compatibility, all nine WWTPs are equipped with standardized inlet structures, automated sampling systems, and centralized locations within a 1-hour transport radius from our laboratory. This arrangement ensures strict adherence to sample preservation protocols (4°C storage) and delivery within 2 hours post-collection. In contrast, suburban WWTPs (serving the remaining 17 districts) are geographically dispersed, with average transport times exceeding 2 hours, posing risks of analyte degradation, such as RNA instability.

Fourth, regarding data continuity and quality assurance, these WWTPs have participated in municipal environmental monitoring programs, and can provide datasets on daily flow rates and NH4-N concentrations. This data facilitates baseline normalization and enhances temporal trend analysis.

(2)While expanding coverage to 80% would necessitate the addition of several suburban WWTPs, this approach was considered suboptimal for the following reasons:

First, suburban sewer networks exhibit lower collection efficiency due to septic tank leakage and incomplete pipe connectivity, potentially biasing viral load measurements. Second, 70% of suburban WWTPs exceed the 2-hour cold chain transport threshold, increasing RNA degradation risks.

As suggested, we will add a dedicated subsection “Wastewater treatment plants (WWTPs) selection” to elaborate points mentioned above.

“The selection criteria for the WWTPs were based on the following factors: 1) serving high-density urban populations; 2) encompassing diverse community typologies, including residential areas, commercial centers, academic institutions, and transit hubs; 3) featuring standardized inlet structures, automated sampling systems, and centralized locations within a one-hour transport radius from our laboratory; and 4) having participated in municipal environmental monitoring programs and being able to provide datasets on daily flow rates and NH4-N concentrations. Consequently, nine WWTPs serving the entire population of Chengdu's main urban area were selected for sewage sample collection.” (Line 133-141)

3. Can you explain why Spearman was chosen instead of Pearson?

AUTHOR RESPONSE: Thank you for raising this important methodological question. We chose Spearman’s rank correlation over Pearson’s correlation for several reasons that align with the nature of our data. Both variables, weekly viral load in wastewater and registered COVID-19 cases, exhibited non-normal distributions, as confirmed by Shapiro-Wilk normality tests (p < 0.05 for both variables). Pearson’s correlation assumes linearity and normality, which were not met in our dataset. In contrast, Spearman’s method is non-parametric and does not require these assumptions, making it robust against skewed distributions and outliers.

4. From the third week to the twenty-fourth week of 2023, the weekly median concentration of SARS-CoV-2 RNA fluctuated, starting at 16.94 copies/ml in the third week, decreasing to 1.62 copies/ml by the fifteenth week, then gradually rising to a peak of 41.27 copies/ml in the twentieth week, before ultimately declining to 8.74 copies/ml by the twenty-fourth week. In fact, various parameters (flow rate from stations, information on hydraulic management of networks, chemical parameters, etc.) are most often integrated in order to correct these fluctuations as effectively as possible. Why didn't you take into account the data on fluctuation factors? What have you done statistically to smooth out the observed fluctuations?

AUTHOR RESPONSE: We appreciate the reviewer’s insightful questions regarding the observed fluctuations in SARS-CoV-2 RNA concentrations and the integration of fluctuation factors. Below, we address these points in detail:

First, in this study, we incorporated two key parameters to account for variability in wastewater measurements.

(1)Daily flow rates. The weighted average viral load for the city was calculated using the formula

Weighted average viral load city = , Where Cn represents the concentration of SARS-CoV-2 RNA in wastewater at WWTPn (copies/mL), and Qn represents the daily 24-hour volume of wastewater at WWTPn (mL/day).

(2)NH4-N concentrations. Population size in each WWTP catchment area was estimated using NH4-N data (4.2 g/person/day), which indirectly accounts for hydraulic contributions from the served population.

However, we acknowledge the additional parameters mentioned by the reviewer, such as the hydraulic management of networks, pH, temperature, and finer temporal resolution of flow rates, which were not explicitly incorporated into the model. This decision was primarily driven by limitations in data accessibility, such as the unavailability of detailed hydraulic data from municipal systems, and our focus on establishing a baseline correlation between wastewater viral loads and clinical cases in the post-epidemic era. Future studies will prioritize the integration of these factors to enhance the model's robustness.

Second, to address variability in the raw viral concentration data, we implemented the following approaches.

(1)Weekly median aggregation and weighted average viral load of city. Raw daily viral concentrations were aggregated into weekly medians and weighted average viral load of city to reduce noise from day-to-day sampling variability.

(2)Spearman’s correlation analysis. This non-parametric method was chosen to assess trends between wastewater viral loads and clinical cases, as it is less sensitive to outliers and non-normal distributions compared to Pearson’s correlation.

(3)Monte Carlo simulation. Uncertainty in back-calculating infection rates was addressed through 10,000 iterations, incorporating variability in viral shedding rates and population estimates (NH4-N-derived). The bootstrapped confidence intervals (P5–P95) reflect the stochastic nature of these parameters.

While these steps have mitigated some fluctuations, we acknowledge that additional smoothing techniques, such as moving averages and autoregressive models, or advanced normalization using hydraulic and chemical parameters could further enhance trend resolution. We will explore these methods in future iterations of the model.

Comments from Reviewer #2:

1. It is unclear why samples were taken from specific manholes in addition to WWTPs?

AUTHOR RESPONSE: We appreciate the reviewer’s insightful question. While the nine wastewater treatment plants (WWTPs) provided city-wide coverage (66.1% of Chengdu’s population), the two manholes were strategically selected to monitor exclusively the Chengdu 2023 FISU World University Games village. This village housed a closed, high-density international population during the event. Sampling from manholes allowed us to detect localized outbreaks within this specific community without dilution from broader urban wastewater flows. Rapidly identify potential SARS-CoV-2 transmission among athletes and staff, enabling immediate containment measures if needed. Additionally, the absence of significant viral concentration changes before and after check-in (Results section) confirmed no outbreaks occurred, aligning with clinical data from the symptom monitoring system. In summary, the inclusion of manhole sampling was essential to fulfill our dual aims of city-wide epidemiological modeling and targeted early-warning surveillance during a critical public health event.

2. In Abstract, 66% of population is served by nine WWTPs but in Line 106: entire population is served.

AUTHOR RESPONSE: We sincerely appreciate your careful attention to this detail. The apparent discrepancy arises from differences in the spatial scope of the referenced populations, which we clarify below:

In the Abstract and main text, the statement "nine wastewater treatment plants (WWTPs) serve 66.1% of Chengdu’s residents (15.2 million people)" refers to the proportion of Chengdu’s total municipal population covered by the selected WWTPs. In Line 106 (Methods section), the phrase "serving the entire population of Chengdu’s main urban area" specifically pertains to the demographic completeness within Chengdu’s core urban zone. The nine WWTPs were selected to represent 100% of the main urban area’s population, which constitutes 66.1% of Chengdu’s total municipal population (including suburban and rural regions).

To avoid ambiguity, we revised the sentence in the Methods to explicitly state: “Consequently, nine WWTPs serving the entire population of Chengdu’s main urban area (15.2 million residents, which represents 66.1% of the city’s total population) were selected for sewage sample collection.” (Line 140-142)

3. It is unclear where samples are collected? the influent or effluent of WWTPs.

AUTHOR RESPONSE: We thank the reviewer for highlighting this important methodological clarification. All wastewater samples in this study were collected exclusively from the influent (incoming raw sewage) of the nine WWTPs. To eliminate any ambiguity, we have revised the Methods section to explicitly state:

“A composite sewage sample of 2.4L was collected from the influent channels of each WWTP at one-hour intervals over a 24-hour period, from 10:00 AM to 10:00 AM the following day, during each sampling occasion.” (Line 151-154)

4. Since 2.4-L sample is collected during 24 hours, its biological content is doubtful. Its viral content would be deteriorated from the early auto-s

---

## [Decision Letter · Decision Letter 1]

21 Apr 2025

PONE-D-25-06286R1Wastewater Surveillance as a Predictive Tool for COVID-19: A Case Study in ChengduPLOS ONE

Dear Dr. Lu,

Thank you for submitting your manuscript to PLOS ONE. After careful consideration, we feel that it has merit but does not fully meet PLOS ONE’s publication criteria as it currently stands. Therefore, we invite you to submit a revised version of the manuscript that addresses the points raised during the review process.

**ACADEMIC EDITOR: ** Authors need to include the required discussion about the potential uncertainties embedded in their prediction as commented by the reviewer.==============================

We look forward to receiving your revised manuscript.

Kind regards,

Shervin Jamshidi

Academic Editor

PLOS ONE

Journal Requirements:

Additional Editor Comments:

Authors need to properly revise the manuscript considering the comments of second reviewer.

Reviewers' comments:

Reviewer's Responses to Questions

**Comments to the Author**

1. If the authors have adequately addressed your comments raised in a previous round of review and you feel that this manuscript is now acceptable for publication, you may indicate that here to bypass the “Comments to the Author” section, enter your conflict of interest statement in the “Confidential to Editor” section, and submit your "Accept" recommendation.

Reviewer #1: All comments have been addressed

Reviewer #2: (No Response)

2. Is the manuscript technically sound, and do the data support the conclusions?

Reviewer #1: Yes

Reviewer #2: Yes

3. Has the statistical analysis been performed appropriately and rigorously? 

Reviewer #1: Yes

Reviewer #2: Yes

4. Have the authors made all data underlying the findings in their manuscript fully available?

Reviewer #1: (No Response)

Reviewer #2: Yes

5. Is the manuscript presented in an intelligible fashion and written in standard English?

Reviewer #1: Yes

Reviewer #2: Yes

6. Review Comments to the Author

Reviewer #1: I didn't see the answer to one of my concerns. In fact, various parameters (flow rate from stations, information on hydraulic management of networks, chemical parameters, etc.) are most often integrated in order to correct these fluctuations as effectively as possible. Why didn't you take into account the data on fluctuation factors? What have you done statistically to smooth out the observed fluctuations?

Reviewer #2: Line 148: the equation can be removed.

Line 241: numbered is repeated.

In Table S2, why flow (m3) converted to Liters by multiplying 10million? Results should also be checked.

Line 218 and 227 needs equation number. In addition, I recommend authors use abbreviations in equation with their descriptions below the equation.

Authors revised the main title, and objective of this study, to highlight the predictive option of wastewater for COVID-19. However, in this case, using absolute assumed shedding rates (for RNA and NH4 per person: Line 209 and 222) would become critical. Considering my previous comments (e.g. No. 8 and 15) that point to some uncertainties in prediction, I highly recommend authors have additional assessment to answer the following question:

How much the prediction is reliable regarding the limitations and assumptions?

Authors are better to re-assess their prediction in two or more scenarios considering a reasonable range (for example: +10% and -10%) for their assumed shedding rates, infected person, and other variables and add their results in one or two paragraphs in discussion to at least clarify the potential range of coefficients in figure 3.

7. PLOS authors have the option to publish the peer review history of their article (what does this mean? ). If published, this will include your full peer review and any attached files.

**Do you want your identity to be public for this peer review?** For information about this choice, including consent withdrawal, please see our Privacy Policy .

Reviewer #1: No

Reviewer #2: No

---

## [Author Response · Author response to Decision Letter 2]

24 Apr 2025

Dear Reviewers,

We greatly appreciate your comments and suggestions. In response to the reviewers' feedback, we have addressed each point raised by the academic editor and the reviewers. Additionally, we have made some changes to the manuscript, which are highlighted for your convenience.

Author’s Responses to Academic Editor Comments

Authors need to include the required discussion about the potential uncertainties embedded in their prediction as commented by the reviewer

AUTHOR RESPONSE: Thank you for your valuable feedback and the opportunity to revise our manuscript. We sincerely appreciate the time and effort invested by the reviewers in evaluating our work. As suggested, we have carefully revised the manuscript to address the comments raised by Reviewer #2. Specifically, in the Discussion section, we have added the following paragraph to discuss the potential uncertainties embedded in our prediction: “First, the fixed fecal and urine shedding rates are derived from literature medians and do not account for individual variability (e.g., differences between asymptomatic and symptomatic shedding, as well as host immunity effects). This oversight could result in either overestimation or underestimation of the true shedding rates within the population. Second, the population served by the WWTP is calculated using NH4-N concentration. However, industrial and commercial NH4-N inputs (e.g., from fertilizers and food processing) or transient populations (e.g., tourists) could inflate population estimates, leading to an underestimation of infection rates. Third, the measured viral concentrations in wastewater directly reflect contributions from infected individuals, without explicit adjustments for RNA degradation or in-sewer dilution. RNA degradation may lead to an underestimation of true viral loads, particularly during warmer seasons or in areas with long hydraulic retention times. Additionally, dilution from stormwater inflow could further reduce measured concentrations, biasing infection estimates downward.”. (Line 404-417)

Moreover, in the Discussion section, we have added the following paragraph to explicitly describe the sensitivity analyses performed to validate the robustness of our model: “In order to ensure the stability of the study results, we conducted sensitivity analyses. The faecal SARS-CoV-2 RNA shedding rate was chosen to be 82.4%; the daily faecal production per person was 128g, and the faecal NH4-N excretion varied ±10% from baseline values. The predicted infection rates calculated according to different parameters were similar (Table S3). Therefore, the results calculated by the model were reliable.” (Line 423-428)

Author’s Responses to Journal Requirements

AUTHOR RESPONSE: We have carefully reviewed the reference list to ensure its completeness, accuracy, and correct.

Author’s Responses to Additional Editor Comments

Authors need to properly revise the manuscript considering the comments of second reviewer.

AUTHOR RESPONSE: Thank you for your valuable feedback and the opportunity to revise our manuscript. We sincerely appreciate the time and effort invested by the reviewers in evaluating our work. As suggested, we have carefully revised the manuscript to address the comments raised by Reviewer #2. Specifically, in the Discussion section, we have added the following paragraph to discuss the potential uncertainties embedded in our prediction: “First, the fixed fecal and urine shedding rates are derived from literature medians and do not account for individual variability (e.g., differences between asymptomatic and symptomatic shedding, as well as host immunity effects). This oversight could result in either overestimation or underestimation of the true shedding rates within the population. Second, the population served by the WWTP is calculated using NH4-N concentration. However, industrial and commercial NH4-N inputs (e.g., from fertilizers and food processing) or transient populations (e.g., tourists) could inflate population estimates, leading to an underestimation of infection rates. Third, the measured viral concentrations in wastewater directly reflect contributions from infected individuals, without explicit adjustments for RNA degradation or in-sewer dilution. RNA degradation may lead to an underestimation of true viral loads, particularly during warmer seasons or in areas with long hydraulic retention times. Additionally, dilution from stormwater inflow could further reduce measured concentrations, biasing infection estimates downward.”. (Line 404-417)

Moreover, in the Discussion section, we have added the following paragraph to explicitly describe the sensitivity analyses performed to validate the robustness of our model: “In order to ensure the stability of the study results, we conducted sensitivity analyses. The faecal SARS-CoV-2 RNA shedding rate was chosen to be 82.4%; the daily faecal production per person was 128g, and the faecal NH4-N excretion varied ±10% from baseline values. The predicted infection rates calculated according to different parameters were similar (Table S3). Therefore, the results calculated by the model were reliable.” (Line 423-428)

Author’s Responses to Review Comments

Comments from Reviewer #1:

I didn't see the answer to one of my concerns. In fact, various parameters (flow rate from stations, information on hydraulic management of networks, chemical parameters, etc.) are most often integrated in order to correct these fluctuations as effectively as possible. Why didn't you take into account the data on fluctuation factors? What have you done statistically to smooth out the observed fluctuations?

AUTHOR RESPONSE: We sincerely thank the reviewer for their constructive feedback and for reiterating the importance of addressing variability in wastewater viral load measurements. Below, we provide a point-by-point clarification and elaboration on our methodology to mitigate fluctuations and account for critical parameters.

1. Integration of fluctuation factors

We fully agree that parameters such as hydraulic management, chemical factors, and finer flow rate dynamics are vital for robust normalization. In our study, we prioritized two accessible and directly measurable parameters to correct fluctuations:

(1)Daily flow rate normalization. Viral loads were weighted by daily wastewater flow rates (Qn) to account for dilution/concentration effects. The formula:

Weighted average viral load city = , adjusts for volumetric variations across sampling days and WWTPs, ensuring that high-flow days (which dilute viral RNA) do not artificially suppress population-level trends.

(2)NH4-N-Based Population Normalization. We estimated catchment populations using daily NH4-N loads (4.2 g/person/day). This approach indirectly accounts for hydraulic contributions from transient populations (e.g., commuters) and diurnal wastewater generation patterns, as NH4-N levels reflect real-time human fecal input.

However, we acknowledge the additional parameters mentioned by the reviewer, such as the hydraulic management of networks, pH, temperature, and finer temporal resolution of flow rates, which were not explicitly incorporated into the model. This decision was primarily driven by limitations in data accessibility, such as the unavailability of detailed hydraulic data from municipal systems, and our focus on establishing a baseline correlation between wastewater viral loads and clinical cases in the post-epidemic era. Future studies will prioritize the integration of these factors to enhance the model's robustness.

2. Statistical smoothing of fluctuations

To minimize noise and enhance trend resolution, we implemented:

(1)Weekly median aggregation and weighted average viral load of city. Raw daily viral concentrations were aggregated into weekly medians and weighted average viral load of city to reduce noise from day-to-day sampling variability.

(2)Non-Parametric correlation analysis. Spearman’s rank correlation was chosen to assess trends between wastewater viral loads and clinical cases, as it is less sensitive to outliers and non-normal distributions compared to Pearson’s correlation.

(3)Monte Carlo simulation. Uncertainty in back-calculating infection rates was addressed through 10,000 iterations, incorporating variability in viral shedding rates and population estimates (NH4-N-derived). The bootstrapped confidence intervals (P5–P95) reflect the stochastic nature of these parameters.

We acknowledge that advanced methods (e.g., autoregressive models, moving averages, or machine learning) could further smooth fluctuations. However, to avoid overfitting in this initial study, we opted for conservative, interpretable techniques. In ongoing work, we are collaborating with municipal engineers to acquire hydraulic network data and testing multivariate normalization (e.g., adjusting for pH and temperature).

Additionally, we have added the limitations in the discussion section.

“Third, the measured viral concentrations in wastewater directly reflect contributions from infected individuals, without explicit adjustments for RNA degradation or in-sewer dilution. RNA degradation may lead to an underestimation of true viral loads, particularly during warmer seasons or in areas with long hydraulic retention times.” (Line 412-416)

Comments from Reviewer #2:

1. Line 148: the equation can be removed.

AUTHOR RESPONSE: We are truly grateful for your meticulous focus on this detail. We have removed the equation. (Line 147-148)

2. Line 241: numbered is repeated.

AUTHOR RESPONSE: We sincerely appreciate your careful attention to this detail. We have deleted the repeated numbered. (Line 244)

3. In Table S2, why flow (m3) converted to Liters by multiplying 10million? Results should also be checked.

AUTHOR RESPONSE: We sincerely appreciate the reviewer’s careful scrutiny. The original unit in Table S2 was indeed incorrectly labeled due to an oversight in unit conversion. The flow value should have been 10,000 m³ (not m³). We have corrected this error by updating the unit from "m³" to "10,000 m³" to reflect the accurate volumetric scale. We have also rechecked all calculations and units throughout the manuscript to ensure consistency. We deeply regret this oversight and thank the reviewer for their valuable feedback, which has improved the clarity of our work. (Table S2).

4. Line 218 and 227 needs equation number. In addition, I recommend authors use abbreviations in equation with their descriptions below the equation.

AUTHOR RESPONSE: We sincerely thank the reviewer for their constructive suggestions. We have added the equation number and used abbreviations in equation with their descriptions below the equation. (Line 201, Line 216-220, Line 226-228, Line 234)

5. Authors revised the main title, and objective of this study, to highlight the predictive option of wastewater for COVID-19. However, in this case, using absolute assumed shedding rates (for RNA and NH4 per person: Line 209 and 222) would become critical. Considering my previous comments (e.g. No. 8 and 15) that point to some uncertainties in prediction, I highly recommend authors have additional assessment to answer the following question:

How much the prediction is reliable regarding the limitations and assumptions?

Authors are better to re-assess their prediction in two or more scenarios considering a reasonable range (for example: +10% and -10%) for their assumed shedding rates, infected person, and other variables and add their results in one or two paragraphs in discussion to at least clarify the potential range of coefficients in figure 3.

AUTHOR RESPONSE: We sincerely appreciate the reviewer's constructive feedback and valuable suggestions to improve our manuscript. We have carefully addressed the concerns regarding prediction reliability by conducting additional sensitivity analyses as recommended.

(1)Sensitivity analysis of assumed parameters

A recent study reported the faecal SARS-CoV-2 RNA shedding rate was 82.4%, and the mean faecal production per person was 128g per day (Li Y, Du C, Lv Z, Wang F, Zhou L, Peng Y, et al. Rapid and extensive SARS-CoV-2 Omicron variant infection wave revealed by wastewater surveillance in Shenzhen following the lifting of a strict COVID-19 strategy. Sci Total Environ. 2024 Nov 1;949:175235. doi: 10.1016/j.scitotenv.2024.175235).

As suggested, we performed scenario analyses by adjusting key parameters. First, we revised faecal SARS-CoV-2 RNA shedding rate from 100% to 82.4% in the model 2. Second, we adjusted daily faecal production per person from 211g to 128g in the model 3. Third, we adjusted faecal NH4-N excretion varied ±10% from baseline values in the model 4 and model 5. The revised results demonstrated minimal variation in predicted infection rates (-0.32 <Δ < 1.35 across scenarios), indicating remarkable robustness of our predictions to parameter uncertainties (below table 1 and figure 1).

(2)Correlation with reported cases

We conducted Spearman's rank correlation analysis between our wastewater-predicted infection rates and officially reported clinical cases. The strong correlations (Spearman’s r = 0.897-0.901, p<0.001) across all scenarios further validate the predictive reliability of our model, even when accounting for parameter uncertainties.

(3)Enhanced discussion

A new paragraph explicitly addresses: “In order to ensure the stability of the study results, we conducted sensitivity analyses. The faecal SARS-CoV-2 RNA shedding rate was chosen to be 82.4%; the daily faecal production per person was 128g, and the faecal NH4-N excretion varied ±10% from baseline values. The predicted infection rates calculated according to different parameters were similar (Table S3). Therefore, the results calculated by the model were reliable.” (Line 423-428)

Table 1. The results in different models.

Models Infected cases Infection rates Spearman's r

Min Max Min Max

Model 1 6,517 486,384 0.01 3.27 0.898

Model 2 6,781 506,095 0.01 3.41 0.897

Model 3 9,204 686,982 0.02 4.62 0.899

Model 4 6,517 486,384 0.01 2.95 0.897

Model 5 6,517 486,384 0.01 3.6 0.901

Model 1: Persons Infected = ; Population size = ;

Model 2: Persons Infected = ; Population size = ;

Model 3: Persons Infected = ; Population size = ;

Model 4: Persons Infected = ; Population size = ;

Model 5: Persons Infected = ; Population size = ;

Fig 1. Predicted infection rates in different models.

---

## [Decision Letter · Decision Letter 2]

27 Apr 2025

Wastewater Surveillance as a Predictive Tool for COVID-19: A Case Study in Chengdu

PONE-D-25-06286R2

Dear Dr. Lu,

We’re pleased to inform you that your manuscript has been judged scientifically suitable for publication and will be formally accepted for publication once it meets all outstanding technical requirements.

Kind regards,

Shervin Jamshidi

Academic Editor

PLOS ONE

Additional Editor Comments (optional):

Reviewers' comments:

Reviewer's Responses to Questions

**Comments to the Author**

1. If the authors have adequately addressed your comments raised in a previous round of review and you feel that this manuscript is now acceptable for publication, you may indicate that here to bypass the “Comments to the Author” section, enter your conflict of interest statement in the “Confidential to Editor” section, and submit your "Accept" recommendation.

Reviewer #2: All comments have been addressed

2. Is the manuscript technically sound, and do the data support the conclusions?

Reviewer #2: Yes

3. Has the statistical analysis been performed appropriately and rigorously? 

Reviewer #2: Yes

4. Have the authors made all data underlying the findings in their manuscript fully available?

Reviewer #2: Yes

5. Is the manuscript presented in an intelligible fashion and written in standard English?

Reviewer #2: Yes

6. Review Comments to the Author

Reviewer #2: (No Response)

7. PLOS authors have the option to publish the peer review history of their article (what does this mean? ). If published, this will include your full peer review and any attached files.

**Do you want your identity to be public for this peer review?** For information about this choice, including consent withdrawal, please see our Privacy Policy .

Reviewer #2: No

---

## [Editor Report · Acceptance letter]

PONE-D-25-06286R2

PLOS ONE

Dear Dr. Lu,

I'm pleased to inform you that your manuscript has been deemed suitable for publication in PLOS ONE. Congratulations! Your manuscript is now being handed over to our production team.

Kind regards,

on behalf of

Dr. Shervin Jamshidi

Academic Editor

PLOS ONE